# Tea and Its Components Prevent Cancer: A Review of the Redox-Related Mechanism

**DOI:** 10.3390/ijms20215249

**Published:** 2019-10-23

**Authors:** Xiangbing Mao, Xiangjun Xiao, Daiwen Chen, Bing Yu, Jun He

**Affiliations:** 1Animal Nutrition Institute, Sichuan Agricultural University, Chengdu 611130, China; kikuxxj1008@163.com (X.X.); dwchen@sicau.edu.cn (D.C.); ybingtian@163.com (B.Y.); hejun8067@163.com (J.H.); 2Key Laboratory of Animal Disease-Resistance Nutrition, Ministry of Education, Chengdu 611130, China; 3Key Laboratory of Animal Disease-Resistance Nutrition and Feed, Ministry of Agriculture and Rural Affairs, Chengdu 611130, China; 4Key Laboratory of Animal Disease-Resistance Nutrition, Chengdu 611130, China

**Keywords:** tea and its components, cancer, ROS homeostasis, anti-oxidative, pro-oxidative

## Abstract

Cancer is a worldwide epidemic and represents a major threat to human health and survival. Reactive oxygen species (ROS) play a dual role in cancer cells, which includes both promoting and inhibiting carcinogenesis. Tea remains one of the most prevalent beverages consumed due in part to its anti- or pro-oxidative properties. The active compounds in tea, particularly tea polyphenols, can directly or indirectly scavenge ROS to reduce oncogenesis and cancerometastasis. Interestingly, the excessive levels of ROS induced by consuming tea could induce programmed cell death (PCD) or non-PCD of cancer cells. On the basis of illustrating the relationship between ROS and cancer, the current review discusses the composition and efficacy of tea including the redox-relative (including anti-oxidative and pro-oxidative activity) mechanisms and their role along with other components in preventing and treating cancer. This information will highlight the basis for the clinical utilization of tea extracts in the prevention or treatment of cancer in the future.

## 1. Introduction

Tea (*Camellia sinensis*) is highly consumed making it one of the most prevalent beverages in the world [1]. There are diverse classifications of the methods used for tea production in different regions. According to the degree of fermentation tea is generally classified into three main types, including the unfermented green tea, the partially fermented oolong tea, and the fully fermented black or pu-erh tea [2]. The functional composition of tea consists of tea polyphenols, tea polysaccharides, *L*-theanine, tea pigments, caffeine, and tea saponin, some of which are secondary metabolites generated by tea. These active chemicals contribute to the many important properties of tea, such as anti-cancer, anti-aging, anti-microbial, anti-inflammatory, hypoglycemic, and hypotensive activities, along with maintaining health and controlling diseases in human [3,4,5,6,7]. As a major compound, polyphenols are rich in tea (about 30%) and possess anti- or pro-oxidative properties that have been studied widely for more than 30 years.

The redox balance is critical for the health of humans. Reactive oxygen species (ROS) are by-products of normal cellular metabolism which have been found to be associated with cancer. Generally, ROS accumulation can cause damage to DNA, proteins, and lipids, and eventually lead to carcinogenesis [8]. However, ROS-mediated oxidative stress may also induce the death of cancer cells [9]. Hence, modulating ROS production may be a potential strategy for cancer therapies. In this review, we will integrate the available information on the relationship between ROS and cancer, and the anti-cancer compounds found in tea, thereby summarizing the potential mechanisms of tea protecting against cancer.

## 2. Cancer

While cancer is a major cause of death little remains definitively known about it. Cancer, also known as malignant tumor or neoplasms, is a family of diseases that involve abnormal cell growth with the potential to invade or spread to other parts of the body [10,11]. However, cancer (particularly cervical cancer) barely produces obvious symptoms in the initial period leading to a late diagnosis and limited treatment options [12]. Cancer is a chronic disease requiring ongoing supports in four key areas, including prevention, surveillance, intervention for consequences of cancer and its treatment, and coordination between specialists and generalist providers [13]. Currently, the modern biomedical science community has accepted a theory that carcinogenesis is attributed to somatic mutation [14]. Somatic cells accumulate mutations in three stages, initially cell survival and proliferation are promoted, followed by the induction of mutations [15]. Thus, gene alterations, especially the activation of oncogenes and the inactivation of cancer suppressor genes, are the main causes of cancer [16,17,18,19,20].

## 3. ROS and Cancer

### 3.1. ROS Homeostasis and Regulation

ROS are a group of highly reactive ions and molecules that are derived from molecular oxygen (O_2_) [21]. The endogenous generation of ROS is primarily derived from the mitochondrial electron transport chain (ETC), a family of membrane-bound NADPH oxidases (NOXs) and 5-Lipoxygenase (5-LOX) [22].

ROS homeostasis is required for normal cell survival and proper cell signaling. Under normal circumstances low levels of ROS can stimulate signaling pathways to regulate intracellular homeostasis and physiological functioning [21,23,24]. However, their fugitive properties and numerous cellular effects potentially make them indiscriminate and lethal oxidants [25]. Irregular ROS accumulation mainly stems from decreasing their elimination and increasing their production. Therefore, in order to guarantee ROS homeostasis, the antioxidant defense system must actively regulate ROS levels.

Enzymatic antioxidants and non-enzymatic antioxidants are the major components of the antioxidant defense system. Enzymatic antioxidants mainly consist of superoxide dismutases (SODs), peroxiredoxins (PRXs), glutathione peroxidases (GPXs), and catalase (CAT) [26,27]. ROS (i.e., O_2_^−^, H_2_O_2_, OH^•^, NO, and ONOO^−^) can be cleaned up and catalyzed to H_2_O by these antioxidant enzymes (Figure 1). The removal of ROS will then inhibit biomolecule damage, including DNA damage, lipid peroxidation, and protein denaturation [8,26,28,29]. Some non-enzymatic materials also play an important role for antioxidation. Glutathione (GSH) is the main non-enzymatic antioxidant in cells. Two molecules of GSH are oxidized by H_2_O_2_ which will produce glutathione disulfide (GSSG), and then GSSG is converted back to GSH by glutathione reductase (GR) and NADPH [30]. In addition, activation of nuclear factor erythroid 2–related factor 2 (Nrf2)-antioxidant response element signaling pathway can stimulate the cellular expression of antioxidant enzymes [31]. The tumor suppressor gene p53 is also implicated in the expression of some important antioxidant genes, and participates in the antioxidant defense system [32]. Thus, Nrf2 and p53 signaling may be considered as the important intracellular pathways for regulating antioxidant capacity.

### 3.2. ROS and Carcinogenesis

Compared with normal cells cancer cells have higher levels of ROS and may maintain a persistent pro-oxidative state, impairing ROS homeostasis, and leading to an intrinsic oxidative stress [33]. Oxidative stress results from the massive accumulation of ROS and may induce carcinogenesis in normal cells [34]. Lipid peroxidation can be induced through ROS reacting with polyunsaturated fatty acids, and its end products includes malonaldehyde (MDA) and 4-hydroxynonenal (HNE) having mutagenicity and tumorigenicity on mammalian cells [35,36]. As a major hallmark lesion 8-oxo-7-hydrodeoxyguanosine (8-oxo-dG) is also generated by ROS reacting with both the DNA and nucleotide pool [37]. Moreover, ROS can specifically hyperactivate the mitogen-activated protein kinase (MAPK) and phosphoinositide 3-kinase (PI3K)/protein kinase B (Akt)/mammalian target of rapamycin (mTOR) signaling pathways, stimulating some intracellular signaling cascades, and resulting in tumor development and metastasis through the regulation of cellular phenotypes including cancer cells survival, proliferation, and angiogenesis [38]. Riemann et al. showed that ROS prevented acidosis-induced MAPK phosphorylation, whereas the addition of H_2_O_2_ enhanced it in AT1 R-3327 prostate carcinoma cells [39]. Reuter et al. and Chetram et al. have reported that the ROS-mediated PTEN function further activating Akt is the hub of complex signaling networks that integrate a multitude of potentially oncogenic signals [40,41].

In addition, ROS act as inflammatory factors which can lead to carcinogenesis. First of all, ROS are involved in the respiratory burst of neutrophils, tumor-associated macrophages (TAM), and lymphocytes, as well as recruiting inflammatory cells into sites of inflammation [42]. Secondly, some studies have revealed that the development of cancer was associated with chronic inflammation [43,44]. The induction of chronic inflammation that predisposes cancers include microbial infections (such as *Helicobacter pylori* for gastric cancer and mucosal lymphoma), autoimmune diseases (such as inflammatory bowel disease for colon cancer), and inflammatory conditions of uncertain origin (such as prostatitis for prostate cancer) [45,46,47]. Zhang et al. also found that esophageal adenocarcinoma cells successfully achieved metastasis and progression through ROS-induced NF-κB signaling pathways, which increased the expression of tumor necrosis factor (TNF)-α, interleukin (IL)-6, and IL-8 [48].

Besides inducing oxidative stress and inflammation, ROS-induced tumor progression and metastatic are also related to the neovascular response [49]. The increased levels of ROS and Akt phosphorylation promote angiogenesis and tube formation [50]. Vascular endothelial growth factor (VEGF) may be up-regulated by oxidative stress, which is known to play a crucial role in tumor angiogenesis [51]. Therefore, decreasing levels of ROS could prevent carcinogenesis and the proliferation of cancer cells.

### 3.3. ROS as a Cancer Therapy Agent

Interestingly, ROS are appreciated for having a dual role in cancer cells. If the generation of ROS exceeds the tolerance range of cancer cells, by using an oxidative stress inducer (such as STA-4783, ampelopsin), this will result in a threat to survival and even the death of cancer cells [9,52]. ROS promotes cancer cell death and autophagy through the activation of c-Jun *N*-terminal kinase (JNK) and p38 signaling pathways [53]. Recently, it has been shown that apoptotic cell death can occur through activating the endoplasmic reticulum (ER) stress/ROS/JNK axis and inhibiting Akt pro-survival signaling in colon cancer cells [54]. Therefore, cancer cells maintain ROS at a level without inducing cell death by promoting pro-tumorigenic signaling pathways and triggering their antioxidant capacity [27] along with controlling the production of mitochondrial ROS (mROS) via cytosolic isocitrate dehydrogenase-1 (IDH1)-dependent reductive carboxylation [55]. These evidences are in opposition to previously published studies. Then what is the true relationship between ROS and cancer? The role of ROS as an oncogenesis agent or a cancer therapy agent is determined by the concentrate and types of ROS, and its local antioxidant capacity [56]. Currently, it is unreliable to use ROS generators as a mono-approach to treat cancer in the clinic and the redox adaptation of cancer cells, which is largely in charge of therapeutic resistance and tumor relapse, needs to be taken into consideration when treating cancer.

## 4. Tea Resists Carcinogenesis

The major tea-producing countries are China, India, Japan, Sri Lanka, Indonesia, and Central African countries [57]. The argument that tea is a cancer preventive agent is no longer new. A pioneering study in the mid-1990s summarized the available epidemiologic information and found that tea consumption is likely to have beneficial effects on reducing the cancer risk in some people [1]. Recently, a meta-analysis found an inverse association between tea consumption and cancer risk [4,58,59,60,61]. However, some evidence does not support the hypothesis that tea can reduce the risk of cancer [62,63]. The above conflicting results could be due to variations in the types, dosage, and drinking manner of tea. In fact, the components and quality of tea are variable by the category, growth environment, storage time, and method of production, which will affect the original beneficial effects of tea.

### 4.1. Tea Polyphenols

Tea polyphenols are one of the most important ingredients in regulating the redox balance of tea. Our previous in vivo and in vitro studies reported that tea polyphenols have strong anti-oxidative capacities [64,65,66]. Tea polyphenols can reduce the incidence and development of tumors in the stomach, intestines, liver, lungs, skin and other parts of the whole body [67,68,69,70,71]. Catechins are the most abundant polyphenols in tea, mainly including epigallocatechin-3-gallate (EGCG), epigallocatechin (EGC), epicatechin-3-gallate (ECG), and epicatechin (EC) [34]. Among them, EGCG is the major catechin in tea, and may account for 50–80% of the total catechins [72].

Tea polyphenols could decrease the risk of skin cancer through inhibiting ultraviolet light B (UVB)-induced oxidative stress, such as the depletion of antioxidant enzymes, lipid oxidation, and the infiltration of inflammatory cells [68]. In a two-stage model of diethylnitrosamine (DEN)/phenobarbital (PB)-induced hepatocarcinogenesis of Sprague-Dawley rats, oral gavage of tea polyphenols five times weekly could significantly increase the total antioxidant capacity (T-AOC) and GPX activity in livers [67]. In the multistage mouse skin carcinogenesis model, peracetylated EGCG treatment could decrease the expression of oxidative enzymes, such as inducible nitric oxide synthase (iNOS) and cyclooxygenase (COX)-2 [73]. In addition, tea polyphenols show a pro-oxidative activity. EGCG-induced oxidative stress and mitochondrial dysfunction, and played an anti-cancer role in oral cancer [74].

In addition to acting as preventive agents, tea polyphenols can also be used as adjuvant therapies for various cancers. When EGCG is combined with a conventional cancer therapy additive, synergistic effects have been proposed, which are mainly due to its anti-inflammatory and anti-oxidative activities that improve the side effects during cancer treatment [75]. Zhang et al. found that the administration of 400 mg EGCC three times daily potentiated the efficacy of radiotherapy in patients [3]. However, EGCG also has antagonistic interactions with PS-341, which will limit its clinical use. PS-341 is an anti-myeloma drug which activity would be blocked by EGCG through vicinal diols on polyphenols interacting with the boronic acid of PS-341 [76]. As a consequence, pre-clinical studies on tea polyphenols (particularly on the bioactive utilization, mechanism of action, and safety of EGCG) need to be carried out.

### 4.2. Others

In the cancer field studies involving tea polysaccharides (TPS), *L*-theanine, tea pigments, and caffeine are not adequate. TPS are a group of hetero-polysaccharides bonded with proteins [77]. Yang et al. reported that TPS (400–800 μg/mL) significantly improved the anti-oxidative capacity in a dose-dependent manner, and inhibited the cancerometastasis of gastric cancer in mice [78]. Selenium (Se)-containing TPS (IC_50_ of 140.1 μg/mL) induced ROS generation which made cells undergo G2/M phase arrest and apoptosis and exhibited effective inhibition of human breast cancer MCF-7 cell growth [79]. Moreover, compared with the utilization of doxorubicin (DOX) alone, a combination of TPS and DOX has a better suppression efficiency in lung cancer A549 cells [80].

*L*-theanine is a natural amino acid which is found specifically in tea plants and makes up 1–2% of the dry weight of tea leaves [81]. Liu et al. found that theanine and its derivates had no toxicity in mice [82]. Recent studies have shown that, in addition to relieving depression, memory improvement, and neuroprotection [83,84,85], *L*-theanine may also have anti-tumor activities. Adriamycin (ADR) was used to efficiently treat Ehrlich ascites carcinoma cells and its side effects, such as reducing antioxidant enzyme activity and increasing the level of lipid peroxidation, can be alleviated by the combined utilization of *L*-theanine [86].

Tea pigments are the oxidized products of polyphenols and their derivatives in tea leaves and mainly consist of theaflavins (TFs), thearubigins (TRs), and theabrownin [87]. The composition of tea pigments in black tea are similar to that of the tea polyphenols in green tea, but the former is chemically stable and may be an ideal chemopreventive agent [88]. In a rat liver precancerous lesion model the treatment with tea pigments suppressed cancer biomarkers such as glutathione S-transferase Pi (GST-Pi) mRNA and protein [89]. Furthermore, in an in vivo trial on 1,2-dimethylhydrazine (DMH)-induced rat colorectal carcinogenesis, treatment with 0.1% tea pigments reduced aberrant cryptic foci (ACF) and colonic tumor formation [90].

Caffeine, the most abundant alkaloid in tea, makes up 2–4% of the dry weight, and its structure is identified as 1,3,7-trimethylxanthine [91]. Caffeine has been shown to have both positive and negative health effects. The cancer preventative effects of caffeine in rodent hepatocellular carcinoma (HCC) models have also been demonstrated [92]. Chronic caffeine ingestion inhibited rat breast cancer, neither by interfering with the high prolactin levels that is a necessary step in murine tumor development, nor by causing hypocaloric intake [93]. However, an in vivo trial showed that the rats consuming caffeine and unsaturated fat had the earliest tumor development and the most multiple tumor occurrence [94].

### 4.3. Tea Types and Anti-Cancer

Along with studying the different components of tea, studies should also be undertaken to analyze their anti-cancer properties. Generally, tea is divided into three main types based on production, namely unfermented green tea, partially fermented oolong tea, and fully fermented black tea or pu-erh tea [2]. We have already summarized and discussed the anti-oxidant capacity of tea polyphenols derived from the differently produced teas but remain unable to draw a consistent conclusion [34]. The individual effects of green tea, black tea, and oolong tea on cancer are difficult to confirm using epidemiological research, mainly due to many consuming several tea types [95]. However, it appears that when comparing the anti-cancer effects between green tea and black tea, the former is more efficient [96,97]. This can be associated with the stronger antioxidant capacity and protective effects of green tea [95,98]. However, Record and Dreosti reported that treatment with black tea provided more protection than green tea in solar irradiation-induced skin cancer in hairless mice [99]. Until now, there has been no direct evidence that oolong tea, a semi-fermented tea, has the ability to fight cancer. Only one in vitro experiment showed that oolong tea has the worst inhibiting effect on the invasion and proliferation of AH109A compared with green and black teas [100].

## 5. Anti-Cancer Mechanisms of Tea through Regulating ROS Homeostasis

### 5.1. Anti-Oxidant Capacity of Tea

Even though a great amount of anti-cancer mechanisms of tea has been discovered the anti-oxidant capacity of tea is still considered as the most important mechanism (Figure 2). Tea polyphenols are the main anti-oxidant capacity component in tea. Tea polyphenols have higher antioxidant capacity than vitamins and offset vitamin’s disadvantages, including photosensitivity from natural vitamin intake and the side effects of synthetic vitamin [101].

#### 5.1.1. Tea as a Direct ROS Scavenger

In an in vitro oxidative hemolysis model of human red blood cells (RBC) green tea polyphenols, as natural antioxidants, efficiently suppressed the hemolysis in the sequence of EGCG > EGC > ECG ≈ EC [102]. Cancer induced by oxidative stress will produce abnormal levels of MDA, HNE, and 8-oxo-dG, which can be mitigated by tea polyphenols [34,103]. Oxidative DNA and protein damage was mitigated by the consumption of tea polyphenols in a mice model with prostate cancer cell subcutaneous xenografts [104]. The decrease of ROS-induced biomolecule damage could depend on EGCG, which is the most effective scavenger for O_2_^−^, OH^•^, and 1,1-diphenyl-2-picrylhydrazyl radicals among tea catechins [105]. EGCG scavenges ROS mainly through oxidizing the B and D ring of the galloyl group, and the oxidation of ECG occurs initially in the D ring [106]. After the phenolic hydroxyl groups of tea polyphenols combine with ROS, phenoxy radical is formed, and then the levels of ROS decrease [107].

Besides tea polyphenols, other components of tea may directly clean up ROS. Theabrownin exerts a ROS scavenging activity because it is the derivative of tea polyphenols [108]. In addition, TPS also can be a direct scavenger to eliminate excessive ROS, but the ROS scavenging activity of TPS depends on the preparation method, drying method, and its concentration [109,110,111].

#### 5.1.2. Tea as an Indirect ROS Scavenger

EGCG may indirectly enhance the antioxidant capacity to suppress carcinogenesis, which is derived from the increasing activities of the antioxidant enzyme and the decreasing effects of the oxidases [103]. TPS can also improve the anti-oxidant function in gastric cancer mice via decreasing the levels of MDA and increasing the activity of antioxidant enzymes [78,112]. Tea extract could attenuate oxidative damage by improving the expression of SOD, CAT, and GPX, and downregulating the expression of iNOS and COX-2 in mice [113]. Further studies showed that EGCG inhibited *iNOS* gene expression, iNOS kinase and COX activities, which would reduce the generation of NO and prostacyclin, and decrease protein and DNA damage, and ultimately inhibit cancer [114,115].

Nrf2/the antioxidant response element (ARE), is the redox-sensitive signaling pathway that regulates antioxidant enzymes and xenobiotic detoxification of enzymes against oxidative stress to maintain the redox balance in normal cells [116]. In addition, multiple protein kinases such as MAPKs, extracellular regulated protein kinases (ERK), and protein kinase C (PKC), were involved in the regulation of Nrf2 transcriptional activity by inducing the phosphorylation of Nrf2 [117,118]. Tea polyphenols pretreatment could modulate the nuclear translocation of Nrf2 by stimulating the ERK1/2 phosphorylation and transcriptionally regulating the expression of antioxidant enzymes downstream (including heme oxygenase (HO)-1 and NQO-1) in HepG2 cells [117]. Gao et al. also reported that EGCG could upregulate the level of HO-1 to improve contrast-induced oxidative stress against carcinogenesis [119]. Interestingly, several studies have indicated that Nrf2 and HO-1 are frequently upregulated in cancer and correlate with a poor prognosis [120,121]. The dark side of the Nrf2/HO-1 axis could be inhibited by the combination of EGCG and metformin, reaching to a level suppressing lung cancer [122].

The ability of tea extracts to scavenge ROS involves the NF-κB signaling pathway [113]. NF-κB also modulates the expression of cancer-associated cytokines such as TNF-α and IL-8 [123]. Following combining with tumor necrosis factor receptor (TNFR), TNF-α stimulates the NOX complex-induced ROS via NF-κB activation and induces carcinogenesis [124,125,126]. IL-8 is a potent neutrophil chemoattractant which can promote the generation of ROS by activating NF-κB [127]. Fabiola et al. and Tomtitchong et al. found that treating gastric epithelial cells with different factors (such as IL-1β and *Helicobacter pylori*) can enhance IL-8 secretion, and tea polyphenols treatment decreased the risk of gastric cancer via inhibition of NF-κB [5,6]. In an analysis report, black tea consumption could mitigate ROS-induced DNA damage, and then prevent carcinogenesis by inhibiting NF-κB transcriptional expression in tobacco users [128].

The above-mentioned information implied that there is a potential cross relationship between Nrf2 and NF-κB that could be regulated by tea and its components during the inhibition of carcinogenesis [113,129]. A study found that the increasing susceptibility of Nrf2 deficient mice to dextran sulfate sodium (DSS)-induced colitis and colorectal cancer was associated with the decreased expression of antioxidant/phase II detoxifying enzymes, in parallel with the enhancement of pro-inflammatory cytokines through stimulating the NF-κB signaling pathways [130]. Furthermore, Li et al. found that tea polyphenols suppressed NF-κB-dependent inflammation, promoted Nrf2 nuclear translocation, and improved the antioxidant capacity during nonalcoholic steatohepatitis in mice fed a high-fat diet [113].

Some drug-metabolizing pathways also are targets of tea and its components. The aryl hydrocarbon receptor (AhR) and cytochrome P450 (CYP450) may convert procarcinogens into carcinogens, which induces cellular toxicity by regulating the generation of ROS [131]. Several studies reported that CYP450 took part in polycyclic aromatic hydrocarbons (PAHs) metabolism, and there is a carcinogenesis possibility causing oxidative DNA damage via ROS generation [132,133,134]. However, green tea and black tea decrease the risk of cancer through inhibiting AhR activation pathways and CYP450 activity in the rat liver [135]. Therefore, it is possible that tea and its components potentially are able to inhibit carcinogenesis through influencing drug-metabolizing pathways to impede ROS production.

Moreover, the antioxidant activity of EGCG reaching anti-carcinogenesis is also attributed to its chelation with metal ions [136]. Iron is required for neoplastic cells and will be prone to depletion [137]. Iron ion supplementation promotes tumor-associated macrophage (TAM) formation because of increased ROS production [138]. But the B ring of tea polyphenols may competitively bind metal ions (including iron), which will decrease the iron intake and ROS level in neoplastic cells [137,139]. Thus, tea polyphenols may reduce ROS generation via competing for metal ions.

### 5.2. Pro-Oxidative Activity of Tea

Pro-oxidative activity tea with its components are considered anti-cancer candidates. They selectively cause programmed cell death (PCD) of cancer cells, but do not damage normal cells (Figure 2). PCD of cancer cells involves hyperactivation of p38, JNK, and p53, which activates the expression of downstream apoptotic proteins (such as Bax, caspase-3, and caspase-9) [53,140,141,142]. In contrast to the anti-oxidative property of tea, EGCG-induced apoptosis or autophagy in cancer cells were due to the significant decrease in mitochondrial membrane potential, which results in mitochondrial dysfunction and increased intracellular ROS [143,144]. Then, as an important mitochondrial redox modulator, sirtuin 3 (*SIRT3*) and its downstream targets (including GSH and SOD) transcription could be inhibited through decreasing the nuclear localization of the estrogen-related receptor α (ERRα) by EGCG in oral cancer cells [145]. Thioredoxin (Trx) and thioredoxin reductase (TrxR), known as antioxidant agents and anti-apoptotic proteins, commonly are overexpressed in the human cancer cells [146]. EGCG treatment may reduce Trx/TrxR via the formation of EGCG-Trx1 (Cys(32)) and EGCG-TrxR (Cys/Sec) conjugates and promote cancer cell death [147]. Human telomerase reverse transcriptase (hTERT) is expressed in over 90% of cancers but not in normal cells [148]. Compared with normal cells, EGCG specifically induced ROS production (especially H_2_O_2_) in cancer cells; and ROS down-regulated *hTERT* expression to promote the apoptosis of cancer cells [149]. Interestingly, Nrf2 frequently overexpresses in human cancer cells and will improve the anti-oxidative activity of cancer cells [150]. However, a combination of EGCG (30 µM) with luteolin (10 µM) efficiently inhibits Nrf2 and synergistically increased apoptosis in the head and neck as well as lung cancer cell lines, including A549 [151].

Tea can also induce the non-programmed cell death of cancer cells. Tea and its components were also prone to have pro-oxidative activity with an excessive concentration of transition metal ions. For examples, tea polyphenols mediated the cellular DNA degradation of ROS-induced lymphoma through the involvement of endogenous copper [152]. Moreover, EGCG induces non-apoptotic death of human cancer cells (both HepG2 and HeLa) via ROS-mediated lysosomal membrane permeabilization [153].

## 6. Conclusions

ROS were appreciated for having a dual role in human cancer including promoting and inhibiting carcinogenesis. Generally, tea and its components act as efficient scavengers of ROS in direct and indirect manners. Interestingly, excessive tea-induced ROS can also cause the PCD or non-PCD of cancer cells. Therefore, tea could potentially be a cancer therapy agent. Moreover, we discussed tea and its bioactive components, especially tea polyphenols, on the basis of their anti- or pro-oxidative activity resisting oncogenesis and cancerometastasis through regulating human redox balance.

In addition, tea and its components may have synergistic effects with conventional anti-cancer measures in clinical practice. However, tea and its components still encounter lots of challenges for clinical application. Usage and dosage of tea and its components, and the various cancer types, may be the prerequisite regulating the bioavailability of tea and its components. Thus, further studies need to be done. For example, how to deliver tea and its components effectively to target sites and protect them from degradation. Furthermore, except for tea polyphenols, other components of tea need more studies in cancer treatment. Thus, we strongly recommend extensive clinical studies of the application of tea and its components with regards to its use as both a preventative and potential therapeutic for cancer.

## Figures and Tables

**Figure 1 ijms-20-05249-f001:**
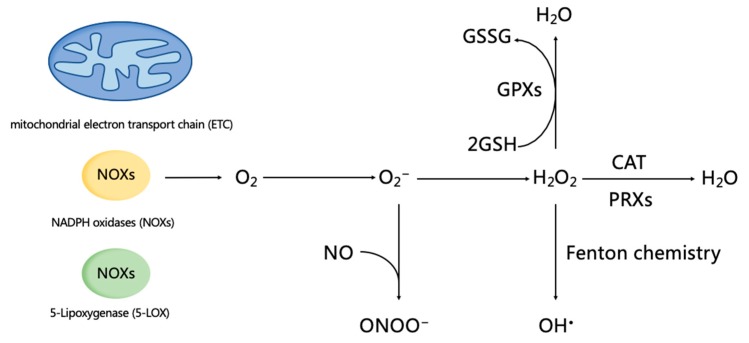
Reactive oxygen species (ROS) generation and homeostasis. The endogenous generation of ROS is primarily derived from the mitochondrial electron transport chain (ETC), including a family of membrane-bound NADPH oxidases (NOXs) and 5-Lipoxygenase (5-LOX). As a precursor of H_2_O_2_ and OH^•^, O_2_^−^can be converted into H_2_O_2_ by three isoforms of superoxide dismutases (SOD) (i.e., SOD1, 2, and 3) that locate in the cytoplasm, mitochondria matrix, and extracellular matrix. O_2_^−^ reacts with nitric oxide (NO), and produces peroxynitrite (ONOO^−^), which will decrease antioxidant capacity. OH^•^ is the production of H_2_O_2_ undergoing Fenton chemistry, and it owns the strongest chemistry activity, which can cause damage to biomolecules (such as DNA damage, lipid peroxidation, and protein denaturation). In addition, H_2_O_2_ can be catalyzed to H_2_O by peroxiredoxins (PRXs), glutathione peroxidases (GPXs), and catalase (CAT). Glutathione (GSH) is the main non-enzymatic antioxidant in cells, and plays an important role in the degradation of H_2_O_2_. Two molecules of GSH are oxidized by H_2_O_2_ through GPX, and are converted into glutathione disulfide (GSSG), and then are regenerated GSH via glutathione reductase (GR) and NADPH.

**Figure 2 ijms-20-05249-f002:**
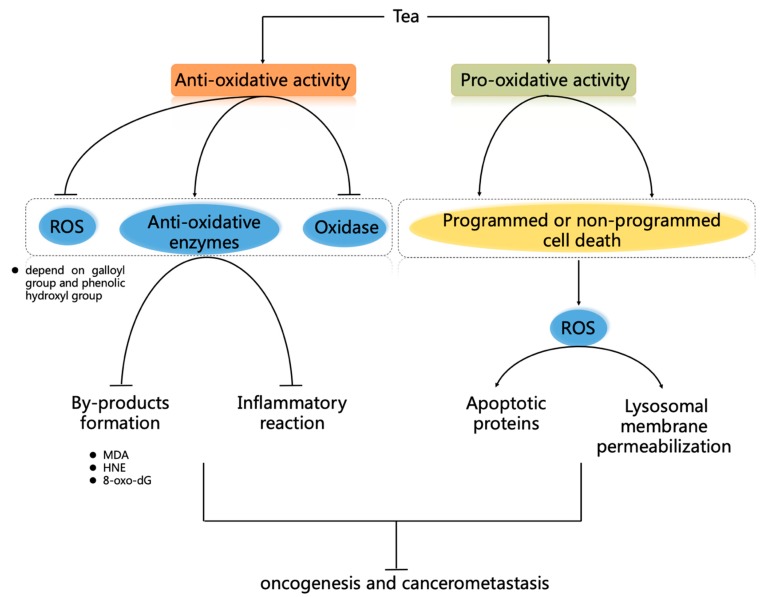
Anti-cancer mechanisms of tea by regulating ROS homeostasis. Anti-oxidative activity of tea. Tea polyphenols can directly scavenge ROS by depending on its B and D ring of the galloyl group, and the phenolic hydroxyl group. Tea can also indirectly eliminate ROS through improving anti-oxidative enzymes activities, decreasing the effects of the oxidases, decreasing by-products (including MDA, HNE, and 8-oxo-dG) formation, and decreasing the inflammatory reaction, which involves the regulation of Nrf2 and NF-κB signaling pathways. Moreover, tea can regulate the drug-metabolizing pathways or become a chelating agent to scavenge ROS. Pro-oxidative activity of tea. With pro-oxidative activity, tea can selectively induce programmed cell death (PCD) or non-PCD of cancer cells, and it is considered as a potential anti-cancer candidate. PCD of cancer cells includes ROS-induced apoptosis or autophagy by mitochondrial dysfunction, the reduction of Trx/TrxR, and ROS-induced hyperactivation of p38, JNK, and p53, which activate the expression of downstream apoptotic proteins (such as Bax, caspase-3, and caspase-9). In addition, tea can induce non-PCD of cancer cells through ROS-induced DNA degradation by endogenous copper and ROS-mediated lysosomal membrane permeabilization.

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
