# Peer review of "Tea and Its Components Prevent Cancer: A Review of the Redox-Related Mechanism"

_ijms, 2019, doi:10.3390/ijms20215249_

Round 1

Reviewer 1 Report

The authors review the redox-relative mechanisms of tea and its components for preventing and treating cancer. The manuscript can be accepted for publication in International Journal of Molecular Sciences, after minor revision. The authors should revise the manuscript according to the following comments.

The sections of "cancer", " ROS and carcinogenesis", and "ROS as a cancer therapy agent" should be limited since these data are common and known. The authors should cite the following articles Toxicol Appl Pharmacol. 2007 Nov 1; 224(3): 265–273. Mol Cells. 2018 Feb 28; 41(2): 73–82. Published online 2018 Jan 31. doi: 10.14348/molcells.2018.2227 Medicines (Basel). 2018 Sep; 5(3): 87. Published online 2018 Aug 10. doi: 10.3390/medicines5030087 Food Science and Human WellnessVolume 2, Issue 1, March 2013, Pages 12-21 There are many grammatical and syntax errors, few examples: Replace duel with dual The sentences should not start with "and" Rephrase the sentence " Reactive oxygen species (ROS), is by-40 products of normal cellular metabolism." rephrase the sentence" People are full of fear to cancer, but little know about it." " their fugitive property" ???

Author Response

Dear Reviewers:

We would like to express our sincere appreciation for your careful reading and invaluable comments to improve this review. We have addressed all issues raised be the reviewer. Special thanks to Miss Qing Yang from Oklahoma State University (USA), and Paul Dyce from Auburn University (USA), for modifying the syntax problem of the overall manuscript. The amendment made are mentioned below with reference to appropriate paragraphs and sections of the revised manuscript.

Response to Reviewer #1

Comments to the Author

The authors review the redox-relative mechanisms of tea and its components for preventing and treating cancer. The manuscript can be accepted for publication in International Journal of Molecular Sciences, after minor revision. The authors should revise the manuscript according to the following comments.

General comments:

Point 1:The sections of "cancer", " ROS and carcinogenesis", and "ROS as a cancer therapy agent" should be limited since these data are common and known. The authors should cite the following articles Toxicol Appl Pharmacol. 2007 Nov 1; 224(3): 265–273. Mol Cells. 2018 Feb 28; 41(2): 73–82. Published online 2018 Jan 31. doi: 10.14348/molcells.2018.2227 Medicines (Basel). 2018 Sep; 5(3): 87. Published online 2018 Aug 10. doi: 10.3390/medicines5030087 Food Science and Human WellnessVolume 2, Issue 1, March 2013, Pages 12-21.

Response 1:Thank you very much for this professional suggestion. According to your requests, we have made extensive modification and cite these articles using at the suitable positions.

Point 2:There are many grammatical and syntax errors, few examples: Replace duel with dual.

Response 2:Thank you very much for this professional suggestion. According to your requests, we have made extensive modification.

Original section of review:

Line 19: Reactive oxygen species (ROS) play a duelrole in cancer cells, including promoting and inhibiting carcinogenesis.

Line 340: ROS were appreciated for having a duelrole in human cancer, including promoting and inhibiting carcinogenesis.

Revised section of review:

Line 19: Reactive oxygen species(ROS) play a dualrole in cancer cells which includes both promoting and inhibiting carcinogenesis.

Line 334: ROS were appreciated for having a dualrole in human cancer including promoting and inhibiting carcinogenesis.

Point 3:The sentences should not start with "and" Rephrase the sentence " Reactive oxygen species (ROS), is by-40 products of normal cellular metabolism."

Response 3:Thank you very much for this professional suggestion. According to your requests, we have made extensive modification.

Original section of review:

Line 40-41: Reactive oxygen species (ROS), is by-products of normal cellular metabolism. Andit has been found to be associated with cancer.

Revised section of review:

Line 42-42: Reactive oxygen species (ROS) are by-products of normal cellular metabolism which have been found to be associated with cancer.

Point 4:rephrase the sentence" People are full of fear to cancer, but little know about it." "

Response 4:Thank you very much for this professional suggestion. According to your requests, we have made extensive modification.

Original section of review:

Line 49: People are full of fear to cancer, but little know about it.

Revised section of review:

Line 51: While cancer is a major cause of death little remains definitively known about it.

Point 5:their fugitive property" ???

Response 5:Thank you very much for this professional suggestion. “their fugitive property” refers to ROS can be eliminated by antioxidants, but some of them also can escape to places away from antioxidants.

Reviewer 2 Report

The review explains the dual mechanisms of Tea in cancer.

some comments need to be addressed before publication,

1- the author should not conclude that Tea can be considered as cancer therapy as many ongoing studies are investigating that in deep.

2- Line 54, there are reappeared word

     Line 55, please delete and

     Line 56, conditions

     Line 71, after the ref. 22 please add comma

     Line 163, please rewrite the sentence.

     Line 220, please add comma after pigments.

3- please check in the whole manuscript the punctuation which disturbed the meaning in many paragraphs.

Author Response

Dear Reviewers:

We would like to express our sincere appreciation for your careful reading and invaluable comments to improve this review. We have addressed all issues raised be the reviewer. Special thanks to Miss Qing Yang from Oklahoma State University (USA), and Paul Dyce from Auburn University (USA), for modifying the syntax problem of the overall manuscript. The amendment made are mentioned below with reference to appropriate paragraphs and sections of the revised manuscript.

Response to Reviewer #2

Comments for the author

The review explains the dual mechanisms of Tea in cancer.

some comments need to be addressed before publication.

General comments:

Point 1:the author should not conclude that Tea can be considered as cancer therapy as many ongoing studies are investigating that in deep.

Response 1:Thank you very much for this professional suggestion. According to your requests, we have made extensive modification in the section 6 “Conclusions”, even all the manuscript.

Original section of review:

Line 352-353:Thus, we strongly recommend extensive clinical studies of the application of tea and its components in all kinds of cancer types.

Revised section of review:

Line 345-346: Thus, we strongly recommend extensive clinical studies of the application of tea and its components with regards to its use as both a preventative and potential therapeutic for cancer.

Point 2:Line 54, there are reappeared word

Line 55, please delete and

Line 56, conditions

Line 71, after the ref. 22 please add comma

Line 163, please rewrite the sentence.

Response 2:Thank you very much for this professional suggestion. According to your requests, we have made extensive modification.

Original section of review:

Line 54: which will lead to late diagnosis and missed the best time of treatment, and then miss the best time of treatments.

Line 55: And cancer is a chronic disease.

Line 56:During survival, cancer patients need to a chronic and complex condition, and require ongoing supports in four key areas, including prevention, surveillance, intervention for consequences of cancer and its treatment, and coordination between specialist and generalist providers.

Line 71: a family of membrane-bound NADPH oxidases (NOXs) and 5-Lipoxygenase (5-LOX)[22]. And there is the mutual conversion among ROS.

Line 163: Catechins, known as the most abundant polyphenols in tea, mainly include epigallocatechin-3-gallate (EGCG), epigallocatechin (EGC), epicatechin-3-gallate (ECG), and epicatechin (EC).

Revised section of review:

Line 55: Cancer is a chronic disease.

Line 56: Cancer is a chronic disease requiring ongoing supports in four key areas including, prevention, surveillance, intervention for consequences of cancer and its treatment, and coordination between specialists and generalist providers

Line 66: a family of membrane-bound NADPH oxidases (NOXs) and 5-Lipoxygenase (5-LOX)[22].

Line 156: Catechins are the most abundant polyphenols in tea, mainly including epigallocatechin-3-gallate (EGCG), epigallocatechin (EGC), epicatechin-3-gallate (ECG), and epicatechin (EC).

Point 3:please check in the whole manuscript the punctuation which disturbed the meaning in many paragraphs.

Response 3:Thank you very much for this professional suggestion. According to your requests, we have made extensive modification.